# Iliski, a software for robust calculation of transfer functions

**Ali-Kemal Aydin**[1,2], **William D. Haselden**[3], **Julie Dang**[2], **Patrick J. Drew**[4], **Serge Charpak**[1,2☯]*, **Davide Boido**[1,5,6☯]*

**1** INSERM U1128, Laboratory of Neurophysiology and New Microscopy, Université de Paris, Paris, France, **2** INSERM, CNRS, Institut de la Vision, Sorbonne Université, Paris, France, **3** Medical Scientist Training Program and Neuroscience Graduate Program, The Pennsylvania State University, University Park, Pennsylvania, United States of America, **4** Departments of Engineering Science and Mechanics, Biomedical Engineering, and Neurosurgery, The Pennsylvania State University, University Park, Pennsylvania, United States America, **5** NeuroSpin, UMR Baobab CEA CNRS, Commissariat à l'Energie Atomique—Saclay Center, Gif-sur-Yvette, France, **6** Université Paris Saclay, Gif-sur-Yvette, France

☯ These authors contributed equally to this work.
* davide.boido@cea.fr (DB); serge.charpak@inserm.fr (SC)

**Data Availability Statement:** Iliski is available on a GitHub repository at https://github.com/alike-aydin/Iliski, along with more example data on Zenodo (http://doi.org/10.5281/zenodo.3773863). The raw data shown in the Figures and used for the

## Abstract

Understanding the relationships between biological processes is paramount to unravel pathophysiological mechanisms. These relationships can be modeled with Transfer Functions (TFs), with no need of *a priori* hypotheses as to the shape of the transfer function. Here we present Iliski, a software dedicated to TFs computation between two signals. It includes different pre-treatment routines and TF computation processes: deconvolution, deterministic and non-deterministic optimization algorithms that are adapted to disparate datasets. We apply Iliski to data on neurovascular coupling, an ensemble of cellular mechanisms that link neuronal activity to local changes of blood flow, highlighting the software benefits and caveats in the computation and evaluation of TFs. We also propose a workflow that will help users to choose the best computation according to the dataset. Iliski is available under the open-source license CC BY 4.0 on GitHub (https://github.com/alike-aydin/Iliski) and can be used on the most common operating systems, either within the MATLAB environment, or as a standalone application.

## Author summary

Iliski is a software helping the user to find the relationship between two sets of data, namely transfer functions. Although transfer functions are widely used in many scientific fields to link two signals, their computation can be tricky due to data features such as multisource noise, or to specific shape requirements imposed by the nature of the signals, e.g. in biological data. Iliski offers a user-friendly graphical interface to ease the computation of transfer functions for both experienced and users with no coding skills. It proposes several signal pre-processing methods and allows rapid testing of different computing approaches, either based on deconvolution or on optimization of multi-parametric functions. This article, combined with a User Manual, provides a detailed description of Iliski

Transfer Functions computations is available in the Supporting Information files as an HDF5 file.

**Funding:** Financial support was provided by the Institut National de la Santé et de la Recherche Médicale (INSERM), the Fondation pour la Recherche Médicale (https://www.frm.org/, EQU201903007811) to SC, the Agence Nationale de la Recherche (https://anr.fr/, ANR/NSF 15-NEUC-0003-02, ANR/TF-fUS-CADASIL and NR-16-RHUS-0004 [RHU TRT_cSVD]) to SC, the Fondation Leducq Transatlantic Networks of Excellence program (https://www.fondationleducq.org/, 16CVD05, Understanding the role of the perivascular space in cerebral small vessel disease) to SC, the IHU FOReSIGHT [ANR-18-IAHU-0001] supported by French state funds managed by the Agence Nationale de la Recherche within the Investissements d'Avenir program to SC and the NIH (https://www.nih.gov/, R01NS078168) to PJD. The funders had no role in study design, data collection and analysis, decision to publish, or preparation of the manuscript.

**Competing interests:** The authors have declared that no competing interests exist.

functionalities and a thorough description of the advantages and drawbacks of each computing method using experimental biological data. In the era of Big Data, scientists strive to find new models for patho-physiological mechanisms, and Iliski fulfils the requirements of rigorous, flexible, and fast data driven hypothesis testing.

This is a *PLOS Computational Biology* Software paper.

## Introduction

Modelling and understanding of the relationship between complex and intermingled biological signals is often difficult, particularly when the drivers of the signals are unknown. The problem of the relationship between two time series can be addressed using deconvolution, which provides Transfer Functions (TFs) representative of the system processing on the input signal to generate the output signal [1]. Extracting the transfer function linking neuronal activity and imaging data is widely used in functional brain imaging [2–9], but TFs can also solve general problems in signal analysis, such as predicting the output of complex electrical circuits [10] or other industrial systems, for which a proper model is overly complex due to multiple processes working in parallel[11]. In brain imaging based on blood flow dynamics, transfer functions are classically used to lump the multitude of cellular and molecular processes linking neural activation to changes in blood flow. This coupling between neural activity and hemodynamics is known as neurovascular coupling (NVC) [12]. While there are many successful phenomenological models of NVC [3,13–17], most physiology-based models of neurovascular coupling[18–21] focus on a single cellular mechanism. As NVC is mediated through multiple processes (several molecular cascades, each mediated by different cell types), a more integrated approach is necessary. NVC has often been assessed with deconvolution [8,22], either in the frequency domain or with matrix-based approaches, like Toeplitz matrices[23]. While these approaches allow the unbiased extraction of the TF, these deconvolution methods suffer from sensitivity to noise, affecting the quality of the computed TFs. Reducing the noise (or bandwidth) of the signals improves the estimate of the TF. Alternatively, one can opt for optimization of known functions or a kernel of functions [7,9]. The first option may lead to information loss, e.g., in cases where the noise is not well characterized. Sophisticated smoothing methods partially prevent this loss, like Savitzky-Golay filter, or noise modelling as proposed by Seghouane and colleagues [24]. The second option relies on parametric functions to find the TF best linking the input to the output signals. The transfer function for neural activity to hemodynamic signals has been canonically modeled using a gamma-distribution function [3,14–16]. While making assumptions as to the shape of the TF has some drawbacks, it is robust in the face of noise and generates parametric representations of intrinsically smooth TFs. These approaches still can suffer from under/overfitting and the search for the minimum of the cost function for ill-posed problems may represent a challenging exercise. A valuable help comes from non-deterministic optimizations like simulated annealing or genetic algorithms, which despite their computational expense have potential advantages in extracting TFs from time series.

Recently, our group has been extensively involved in TF computation of neurovascular coupling in a study based on multi-modal recordings, namely two-photon microscopy and ultra-fast functional ultrasound [25]. For the required task, we comprehensively tested many

deconvolution and optimization algorithms to choose the best-suited approach. We noticed that there is no standard software package providing all these different TF extraction tools, nor a program where all these approaches are available in a comfortable signal pre-treatment and I/O workflow. Here, we present Iliski ([ɪlɪʃkɪ], meaning "relationship" in Turkish), a software which contains all the functionalities that we previously used (Aydin et al.) and which, being open source, can be further improved by the users. Although Iliski was initially thought to help data analysis in Biology, its features make it suitable for diverse applications [26,27].

## Design and implementation

Iliski can compute TFs between an input and an output time series, regardless of their nature. The originality of Iliski resides in its multiple options to process and analyze input signals. Iliski provides users with efficient pre-treatment and several deconvolution or optimization algorithms, through a clear graphic interface. It is meant to be easy-to-use for anyone, even with basic digital signal processing skills. The experienced users, instead, will find both convolution and function optimization approaches–two classes of problems usually comprised in different toolboxes—in a single data analysis environment.

Iliski can be used either as a suite of functions or through a Graphical User Interface (**Fig 1A**). Functions are grouped according to the analysis workflow to keep the interface simple. **Fig 1B** shows the general purpose of Iliski.

### Data loading and pre-treatment

We propose two input files format: either plain text files or HDF5 data, the latter being an open-source file format with advanced database features. As experimental acquisitions are prone to multiple component noise, we provided, as an option to the analysis workflow, smoothing (Savitzky-Golay method) and median filter functions, to exclude outliers. The input and output signals are interpolated to a chosen time interval ($\Delta t$). Both signals can be cut between two given time points to study continuous recordings while computing TFs on chunks of signal (**Fig 1C**). As an option, boxcar function of variable duration can be used in place of the input signal. Note that Iliski was not coded to handle complex-valued signals.

### TF computation options

Two main types of TF computation are proposed: deconvolution or function optimization. The former is straightforward, either Toeplitz or Fourier deconvolution, and does not require any specific settings. The latter is the optimization of a parametric function, which requires further settings depending on the chosen algorithm. Beside the proposed fitting functions, the users can input their own function in the graphical interface or add it to the default ones by modifying a text file (the procedure is described in the Iliski Manual). The TF dimension is user-defined, with setting of the TF duration and 'Sampling Time' parameters that match the original data or can be augmented by non-linear interpolation. Optimization of parameters can be done with various Matlab algorithms, each coming with pros and cons (see Results section) (**Fig 1C, middle**).

### Evaluation of the TF accuracy

A TF is evaluated comparing its prediction–the convolution of the input signal and the TF—to the expected output. Two metrics are used in Iliski: the Pearson coefficient (*corrcoef* function, Matlab, **Fig 1C, right**) and the residual sum of squares. The former was chosen to have a metric solely focusing on the dynamic, allowing for inter-subject comparisons, while the latter

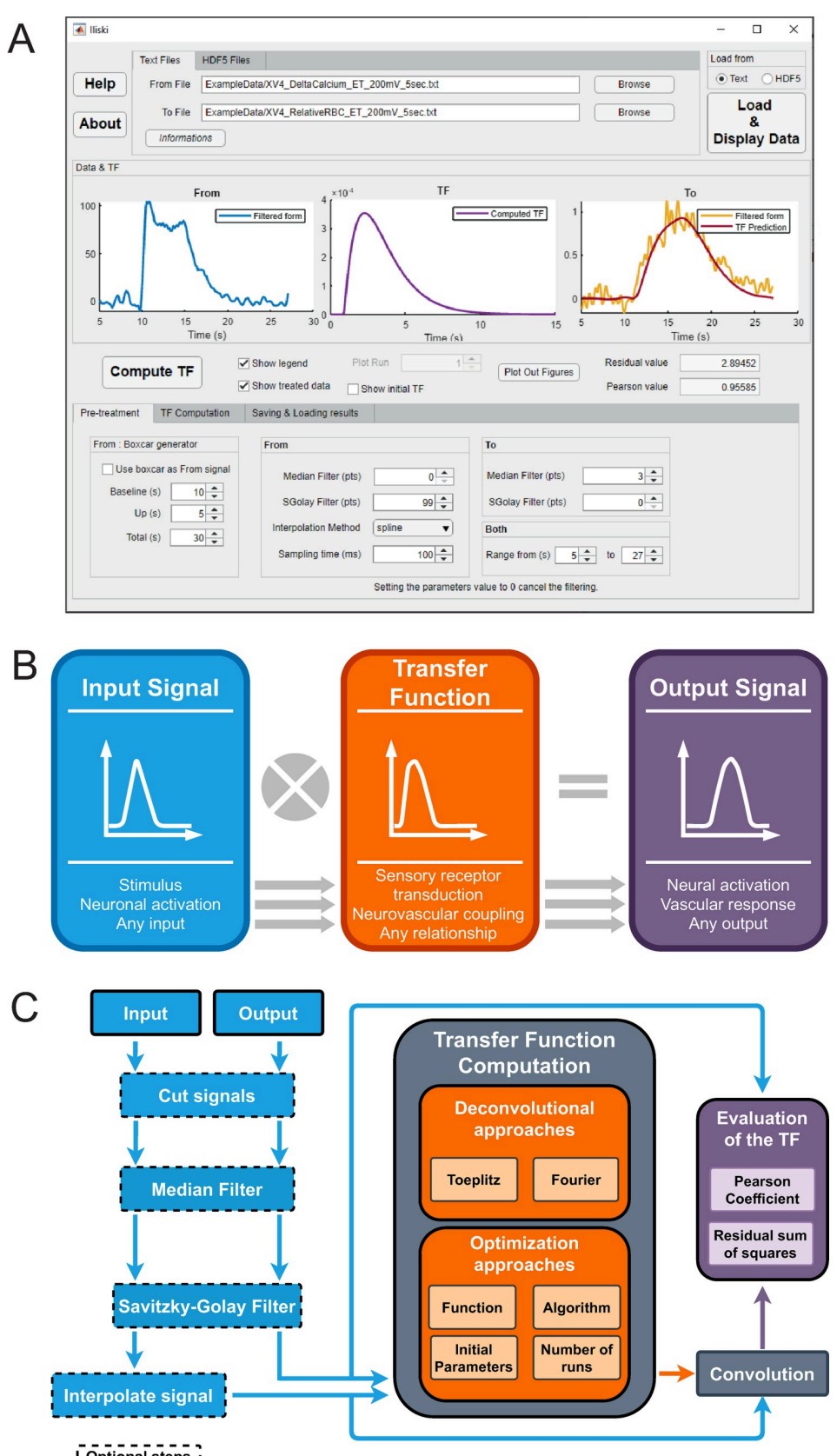

**Fig 1. Overview of Iliski.** (A) Iliski has a clear interface with tabs bringing through the analysis steps. (B) The usage of Iliski are many; although it has been conceived for biological data, there is no limitation to load any discretized signal. Iliski can easily be used as a tool for fast testing different approaches for TFs computation. (C) Iliski workflow is modular so that signal pre-processing is optional and functions to compute TFs can be modified by the user preserving the I/O modules.

evaluates the overall fit, considering the amplitude of the prediction. The cost function of all the optimization algorithms tested in this article is the residual sum of squares (**hereinafter referred to as "residuals"**).

## Post-computation

The results structure is arranged to be as informative as possible while avoiding useless repetition of data. Iliski allows for loading previously computed results structures to check them again. After TF computation, results structure can be saved either as XLS file, readable by any Excel-like software, or as a MAT-file (MATLAB formatted binary file format), but it is also available in Matlab workspace to be exported in various data formats by the user.

## Implementation

Iliski is accessible both as a GUI and as a set of functions to be used in scripts. It has been developed using Matlab R2018a, with the following dependencies: Optimization Toolbox, Signal Processing Toolbox and Global Optimization Toolbox.

Common user errors are thoroughly prevented by various messages and fail safes. In parallel, all errors are treated and saved in a log file, to allow for efficient bug-fixing by any developer. We purposely kept just a few parameters to modify through the GUI, with the goal of providing an easy-to-use tool for people not used to these functions. In most cases, Matlab default parameters of each deconvolution/optimization function worked well with our data, and we believe that it can be extended to many biological datasets. However, a user skilled with Matlab and optimization algorithms can easily modify the parameters of each function used.

## Animal research

This study uses already published data of animal experimentation (Aydin et al.). All animal care and experimentations were performed in accordance with the INSERM Animal Care and Use Committee guidelines (protocol numbers CEEA34.SC.122.12 and CEEA34.SC.123.12).

## Results

Here we present the use of Iliski to find the best mathematical representation of neurovascular coupling, an ensemble of cellular mechanisms that links brain activation to local increases of blood flow. Neural activity is reported by GCaMP6f [28], a calcium-sensitive protein expressed in specific neurons. Blood flow is quantified by measuring red blood cells velocity changes in capillaries [29].

Several deconvolution and function optimization algorithms are provided. Choosing the algorithm(s) and settings to compute a TF that gives faithful and robust predictions is not always a straightforward task. It must be done according to the data features. Here we use some data from our published study on neurovascular coupling [25] to point out how TFs change with different algorithms and settings, and we show the critical points in the usage of non-deterministic methods. Finally, we propose a step-by-step guide to optimize the best TF on practical situations.

## Choosing the best TF computation approach

Fig 2 shows TF computation with different settings over the same couple of signals: neuronal ($Ca^{2+}$) activations and vascular (red blood cells velocity) flow increases recorded in a mouse upon odor application. Our example data display unavoidable and complex noise coming from many sources: the biological system, the optical setup, the electronics, etc. Deconvolution with Fourier or Toeplitz approaches predicts the vascular responses very well for a given data set. However, the high-frequency noise is amplified by deconvolution [24] and transmitted to the TF, the predictions are not robust across data sets and the actual dynamics of neurovascular coupling is completely hidden in the TF noise (Fig 2). In this example, we show what we regard as a typical case of overfitting. The TF is capturing the high frequency noise of the system because it does not have any previous expectations for the shape of the relationship between the input and the output signals. This contrasts to the optimization of a parametric function approach which, although it imposes constraints on the shape of the TF, gives meaningful neurovascular relationship and does not need noise clearing. In blood flow-based neuroimaging, the standard function used to represent neurovascular coupling is composed of one or two $\Gamma$ functions, depending on the nature of the signals, i.e. purely vascular or based on oxygen level[30].

Below is the one $\Gamma$-driven function we used with our data.

$$\text{TF}(t; p_1, p_2, p_3, p_4) = H(t - p_3) \cdot p_4 \cdot \frac{(t - p_3)^{p_1 - 1} \cdot p_2^{p_1} \cdot e^{-p_2 \cdot (t - p_3)}}{\Gamma(p_1)}$$

Where $p_1, .., p_4$ are the parameters to optimize, and $H$ is the Heaviside function that includes a time-shift parameter ($p_3$), In some cases, the time shift significantly improved the prediction and is a known biological phenomenon to consider[31]. Its four parameters are not all independent from one another, e.g., $p_1$, $p_2$ and $p_4$ all impact the TF amplitude. This inter-dependency between the parameters brings an ill-posed optimization problem with multiple local minima of the cost function, the sum of the residual squares, in the 4D space of the parameters. To tackle the function optimization problem, we chose standard algorithms (all implemented in the Optimization Toolbox of Matlab) to encompass the main available options.

A derivative-free optimization method [6] is provided by the *fminsearch* function in Matlab, which uses the Nelder-Mead simplex algorithm. This approach on our data produced a TF with more than one underivable point that is not representative of the smooth dynamic of neurovascular coupling.

Another common option is provided by Quasi-Newton optimization algorithms, which uses an approximation of the Jacobian: for this approach too, we tested an unconstrained built-in method (*fminunc* function, Matlab). This prediction is, overall, as good as with *fminsearch* (Fig 2, Pearson coefficients, *fminunc* vs. *fminsearch*: 0.95 vs. 0.96), but the onset phase is not properly fit. Moreover, although not evident from the plot, the optimized time shift was negative (-120 ms), implying that the onset of the vascular response *precedes* the neuronal activation.

All the optimization methods tested above are deterministic, meaning that repeating them with the same initial parameters will bring the same result. The pitfall of these methods when applied to ill-posed problem is that optimization process will get attracted to the nearest local minimum, regardless of the many other deeper minima, which may be far away in the parameters space. In other words, deterministic algorithms are sensitive to the initial parameters set before starting the optimization.

Non-deterministic algorithms exist to overcome the local minimum issue, adding some level of randomness in the optimization process, and for this purpose Iliski uses the Simulated

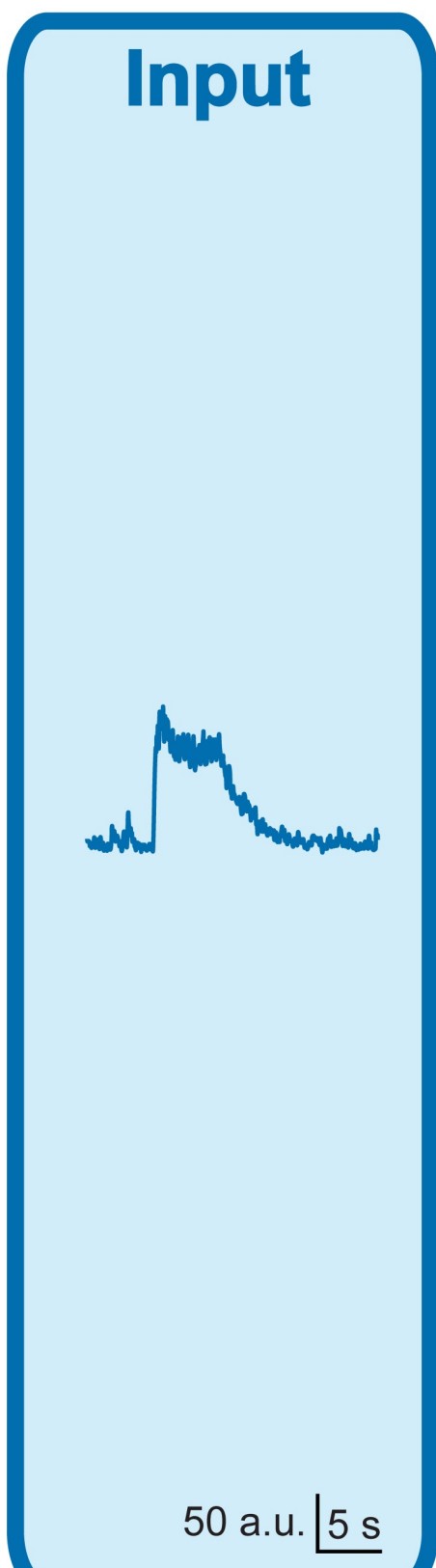
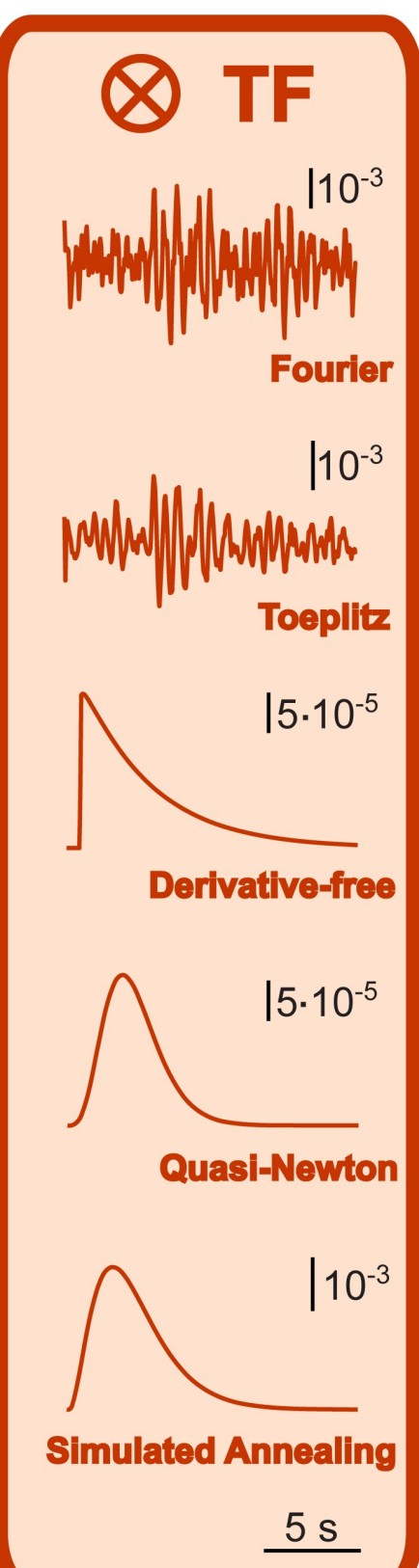
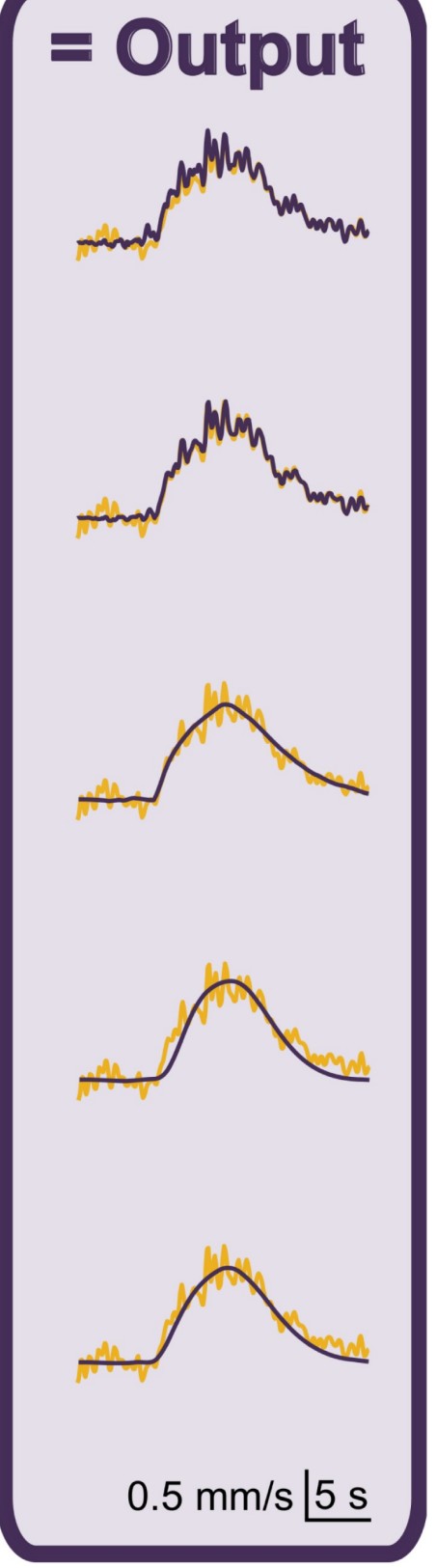

**Fig 2. Comparison of deconvolution and optimization algorithms on a batch of data.** Odor stimulation elicited a neuronal response in the Olfactory Bulb of a mouse, reported by a calcium-dependent fluorescent signal (in blue, left panel), providing the input of TF computation. Output is given by the vascular response, measured as the change in speed of red blood cells flowing inside a capillary proximal to the recorded neuronal activation (in yellow, right panel). Both experimental data have been resampled at 50ms and used to compute a set of TFs (in orange) either with direct deconvolution approaches (Fourier or Toeplitz methods, middle-upper panel TFs) or with 1-Γ function optimization performed by 3 different algorithms (middle-lower panel TFs). Complex TFs bring accurate prediction but amplify the noise of the data used to deconvolve them, with a consequent loss of robustness on other datasets. Smoother TFs are less accurate on the training dataset, but much robust when applied to test datasets.

**Annealing algorithm.** Each optimization run can yield a different result, reaching possibly a different cost function's minimum each time. We define as 'run' a single application of the optimization with a given set of initial values, and 'iteration' the ensemble of runs sharing the same initial values. By running the algorithm multiple times, one can choose the result with the lowest residual, while avoiding TFs which shape are biologically not acceptable. In fact, to represent the NVC, a TF cannot start at 0 sec, because of the delay due to the cellular cascades triggering the vascular response, and it must be smooth to comply with the progression of biological processes. In Aydin et al. (2020), we described a workflow of runs and iterations to get to biologically consistent TFs (see Supplementary Fig 1 in Aydin et al. [25]). To speed up computation, we imposed bounds over the parameters. Note that such bounds can be set through the Iliski GUI for any constrainable algorithm.

Using our data, Simulated Annealing gave a smooth TF and a prediction as good as *fminsearch* for the onset phase of the vascular response. The data shown in **Fig 2** is representative of the rest of the data. In fact, optimization of TFs using neural and vascular recordings from other mice, tested with the same odor stimulation, produced similar residual values of the cost function across the 3 optimization algorithms presented above (1-way ANOVA, $F_{(2, 17)}$ = 0.035, p = 0.97, **Fig 3A**). However, as in the example of **Fig 2**, deterministic algorithms are prone to biologically inconsistent TFs (**Fig 3B**). The non-deterministic, Simulated Annealing algorithm with subsequent iterations method allows to efficiently exclude these TFs and obtain the best trade-off between prediction performance and biological consistency at the cost of a longer computation time. Direct deconvolution is a good option when the goal is the prediction quality within the training database. Deterministic optimization algorithms are fast but yield to TFs that may have biologically inconsistent dynamics. Note that for all the computations we used a short $\Delta t$ (50 ms) for interpolation to preserve most of the information.

## Evaluating the number of runs in a non-deterministic case

As already mentioned, the Simulated Annealing algorithm requires several runs and iterations to obtain a good TF, where 'run' means a single optimization and 'iteration' an ensemble of runs sharing the same initial values of the fitting function. In our experience, starting the optimization with a 'bad' TF—whose shape is different from what is expected for the processed dataset—helps to collect more local minima in a pool of optimization runs. For example, in our previous study [25], we proposed iterations of 50 runs and started with the initial values of the standard TF (one Γ HRF) which, peaking at 5 seconds, turned to be much slower than any of the optimized TFs. The sequence of 50-runs iterations stopped when, within an iteration, no clear improvement was found in the optimized TF [25]. On average, 2 iterations were sufficient to get a stable TF with Pearson coefficient above 0.9. Here, we investigated if a higher number of runs is beneficial to the detection of the minimum of the cost function and if it prevents the need for further iterations. We compared 50 and 200 runs with single and double iterations, in cascade (**Fig 4A**). In a mouse dataset, we observed a non-significant trend towards more scattered TFs shapes for computation using 50 runs versus 200 runs (1-way ANOVA, $F_{(3, 16)}$ = 2.086, p = 0.14, F**ig 4B**). Similarly, the quality of the TFs did not

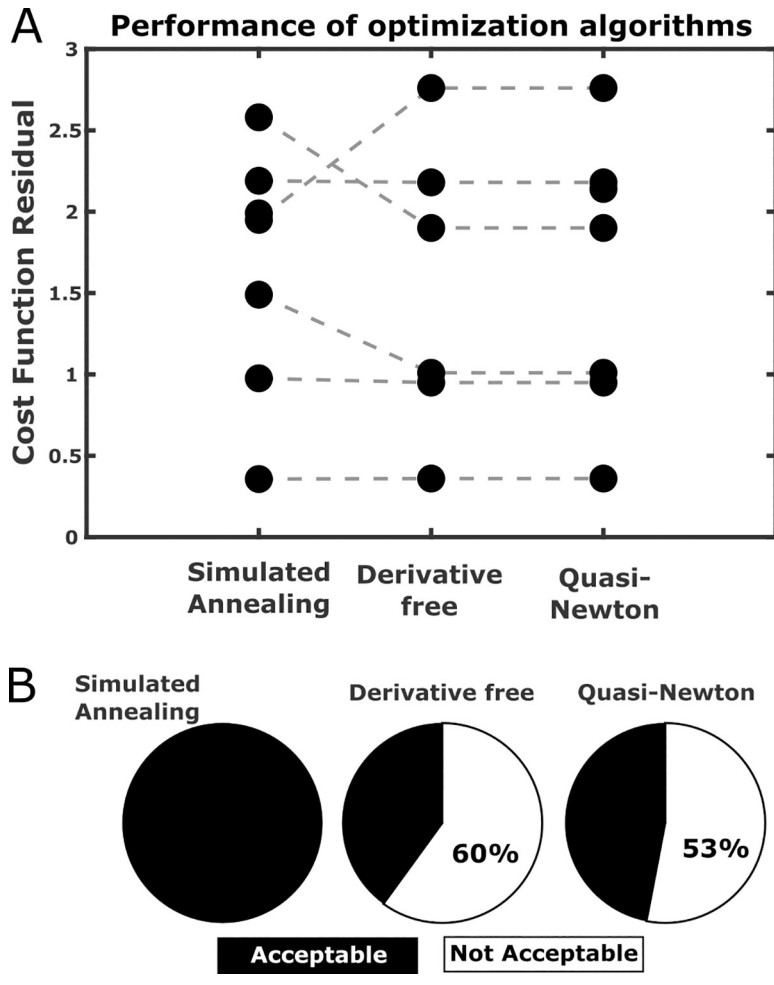

**Fig 3. Prediction performance of different optimization algorithms.** (A) 3 algorithms were compared in terms of the residuals of the cost function of the optimized TF on 7 mice datasets (Derivative free algorithm failed in optimizing a TF in a mouse). No significant difference was found across the 3 methods. (B) However, simulated annealing was the only approach to provide TFs consistent with the nature of biological data (TF with no more than 1 non-derivable point), while both the other deterministic methods run into inconsistent TFs in roughly 60% of the cases.

significantly improve with increasing runs (1-way ANOVA, $F_{(3, 16)}$ = 2.299, p = 0.12). As a result, TFs with fast dynamics (peaking within 1 and 2 sec), was a common feature independently of the adopted protocol (**Fig 4C**). In a dataset from another mouse (**Fig 4D and 4E**), TFs with sparse time to peak values after 200 runs improved after a second iteration, with the same number of runs (2.3 ± 0.3 s VS 1.5 ± 0.1 s (mean ± SEM), two-tailed T-test, unpaired, p = 0.02 < 0.05). Note that this compression of TF dynamics was not accompanied by a significant improvement of the TF quality (residuals: 10.1 ± 2.1 vs. 6.9 ± 0.8 (mean ± SEM) for 200 and 200 + 200 runs respectively, two-tailed T-test, unpaired, p = 0.19). To conclude, depending on the input/output signals, non-deterministic algorithms can produce TFs with different dynamics but close performances in the prediction. The choice of a specific optimization process, with more or less iterations and runs, becomes crucial when the interest is not limited to the prediction quality, but extends to the temporal dynamics of the TF. Because of the noise, TFs with distinct shapes can yield very close residual values.

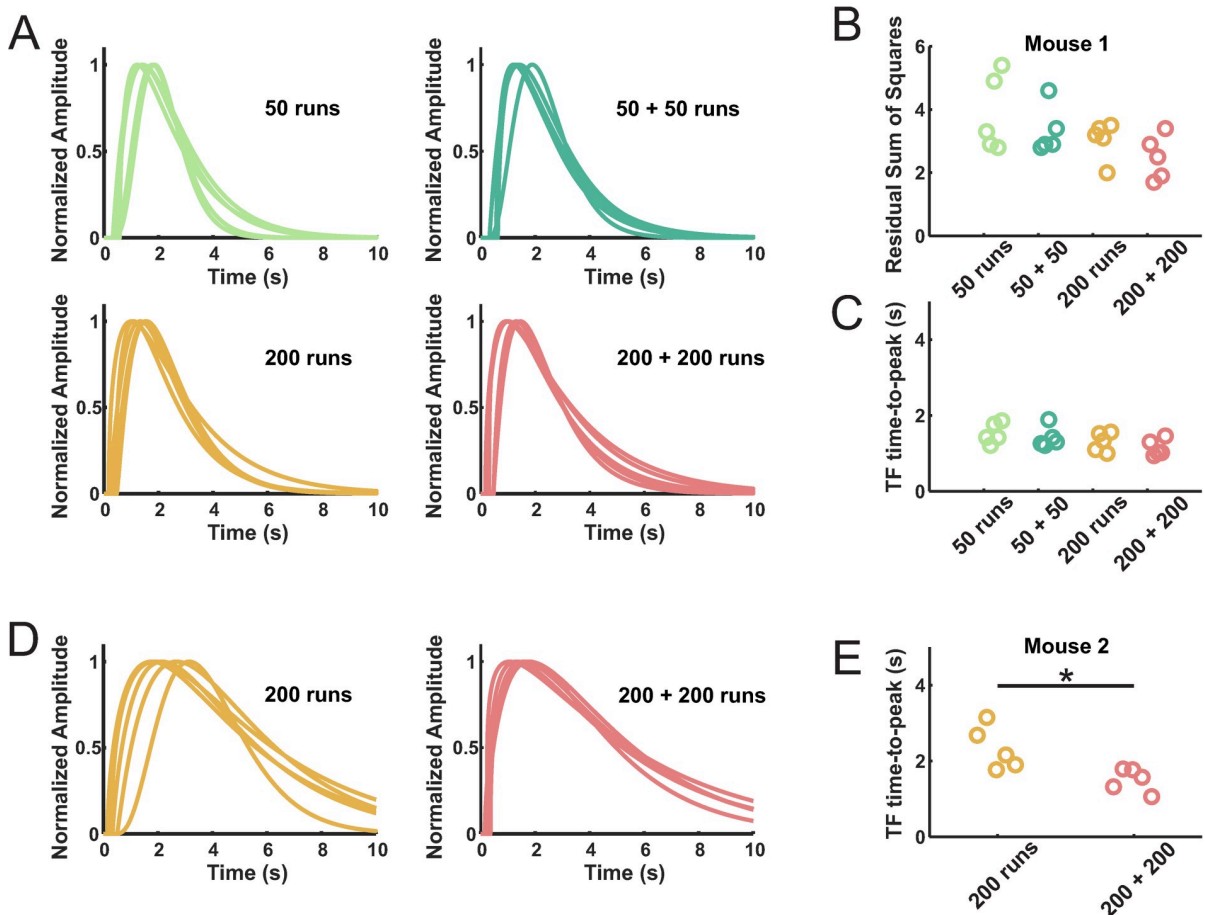

**Fig 4. Influence of the number of runs and iterations on the TF shape and quality.** (A) Using the Simulated Annealing algorithm, we tested 4 protocols of 50 or 200 optimization runs, either done a single time or repeated (5 TFs computed for each protocol). (B) Residuals of the cost function do not significantly differ across the protocols, although the protocols with the highest number or runs show a trend of smaller residuals. (C) Similarly, there was no significant difference for TFs time-to-peak values. (D, left) Same protocols comparison on a dataset from a different mouse revealed a sparse dynamic of optimized TFs, even if the best TFs were selected on a pool of many TFs (200 runs). (D, right) A second iteration of 200 runs gave more homogeneous TFs dynamics. (E) Quantification of the dynamic heterogeneity was made by measuring the time-to-peak which resulted in a scattered distribution for the 200 runs protocol, packed up repeating the same iteration a second time. Residual values, not reported in this figure, were not significantly different for mouse 1 and 2.

## Guide to choose the algorithm best fitting your needs

We provide a decision diagram to choose the best approach to compute a TF based on the features of the user's dataset (**Fig 5**). Nonetheless, we believe it is always a good choice to test different approaches before making the final choice.

## Discussion

Iliski provides a user friendly, interactive, and rich in options software for quickly testing different methods and settings to compute TFs between biological processes. In addition to its standard integrated functions, it also allows for user-defined functions of any number of parameters and the possibility of replacing an input signal with a boxcar function, enlarges its usage. Using data from the NVC field, we demonstrate how critical is the choice of the method for computing TFs and the caveats of parameters such as the number

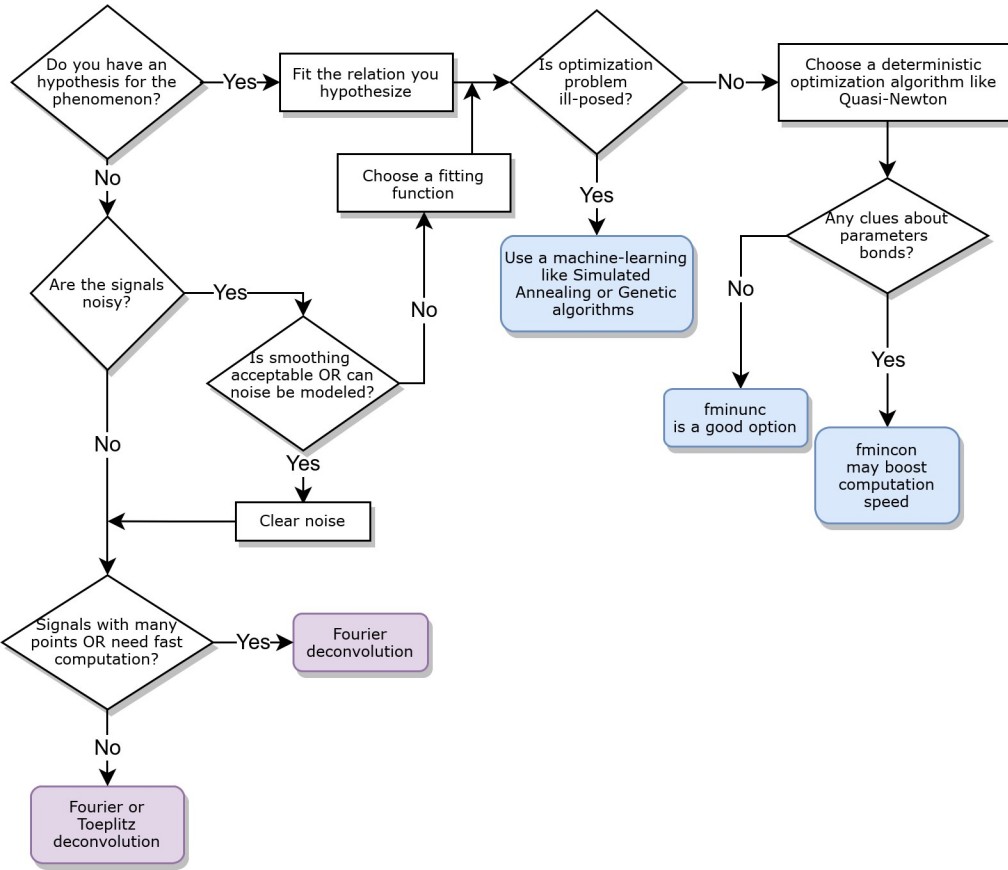

**Fig 5. Decision tree to help choosing the most efficient method to compute a TF with Iliski, based on the data features.**

of iterations necessary to non-deterministic algorithms. Note that we did not report the influence of smoothing, interpolation, fitting and cost functions choice that are also known to affect the result. The use of multimodal datasets, i.e., neuronal calcium signal, measurements of vascular responses at both the microscopic and mesoscopic scales enabled us to demonstrate that NVC is represented by a similar TF which is much faster than the classical HRF, a finding which is getting accepted in the field of brain imaging based on blood flow [5,25,32,33].

## Availability and future directions

Iliski is open-source and freely available under the Creative Commons Attribution 4.0 International (CC BY 4.0) license. Iliski is maintained on GitLab, enabling user-friendly bug report and community work to make the tool fit the users' need. It can be found here: https://github.com/alike-aydin/Iliski.

In the neurovascular imaging field, computing the hemodynamic response function is paramount to interpreting vascular activation in terms of neural activation. In any other field, computing TFs may be of help to go deeper in the interpretation of the results. For these reasons, we think it is extremely helpful to have a data analysis tool which lets fast testing of different algorithms with a user-friendly interface.

## Supporting information

**S1 Data. Raw Data: This HDF5 files contains both the raw numerical data shown in the figures and the experimental biological data used to compute them with Iliski.**
(H5)

**S2 Data. GitHub Repository Clone: This ZIP file contains a clone of the GitHub repository at the time of the publication (also available at https://doi.org/10.5281/zenodo.4765555).**
Current version is available at https://github.com/alike-aydin/Iliski.
(ZIP)

**S3 Data. Example Data: This ZIP file contains example data for testing purposes.** A detailed description of the data is available on Zenodo at http://doi.org/10.5281/zenodo.3773863.
(ZIP)

**S1 Text. User Manual: This is Iliski's User Manual at the time of the publication.** Current version is available in the GitHub repository at https://github.com/alike-aydin/Iliski.
(PDF)

## Acknowledgments

We thank Yannick Goulam Houssen for his insights during the development process.

## Author Contributions

**Conceptualization:** Ali-Kemal Aydin, Serge Charpak, Davide Boido.

**Data curation:** Ali-Kemal Aydin.

**Formal analysis:** Ali-Kemal Aydin, Julie Dang, Davide Boido.

**Funding acquisition:** Serge Charpak.

**Investigation:** Ali-Kemal Aydin, Davide Boido.

**Methodology:** Ali-Kemal Aydin, William D. Haselden, Davide Boido.

**Software:** Ali-Kemal Aydin, Julie Dang, Davide Boido.

**Supervision:** Serge Charpak, Davide Boido.

**Validation:** Ali-Kemal Aydin.

**Visualization:** Ali-Kemal Aydin.

**Writing – original draft:** Ali-Kemal Aydin, Davide Boido.

**Writing – review & editing:** Ali-Kemal Aydin, William D. Haselden, Patrick J. Drew, Serge Charpak, Davide Boido.

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
