## [Decision Letter · Decision Letter 0]

12 Feb 2021

Dear Dr charpak,

Thank you very much for submitting your manuscript "Iliski, a software for robust calculation of transfer functions" for consideration at PLOS Computational Biology. As with all papers reviewed by the journal, your manuscript was reviewed by members of the editorial board and by several independent reviewers. The reviewers appreciated the attention to an important topic. Based on the reviews, we are likely to accept this manuscript for publication, providing that you modify the manuscript according to the review recommendations.

Sincerely,

Dina Schneidman

Software Editor

PLOS Computational Biology

[LINK]

Reviewer's Responses to Questions

**Comments to the Authors:**

Reviewer #1: The manuscript presents the software toolbox Iliski; a user-friendly toolbox designed to determine transfer functions between input and output signals.

The determination of transfer functions is elementary and has applications in many fields including biology. Therefore, an easy-to-use program with hints on pitfalls is a great benefit for the community. The program can be operated intuitively. A comprehensive manual is provided. The program reads the provided test data, as well as data from our own lab. Error messages are a little sparse.

The manuscript might benefit, if the authors mention other possible applications more prominently, such as optical impulse response function of biological tissue (DOI: 10.1109/IEMBS.1996.651999) or the testosterone regulation system (ISBN 978-91-554-8857-4). Also in other fields as electronic processing, acoustic or economics transfer functions are elementary.

The authors do not compare Iliski to any other program for transfer function determination. Is there really no other program for this purpose in such an elementary field.

Specific points:

line 18: The term ‚biological event‘ is not appropriately used. The point is to link related inputs and outputs. An event describes only one time point. Furthermore, the toolbox can be used for all kinds of systems and not only in biology.

line 37: The end of the sentence does not fit the beginning: The TF of neurons is not used for the prediction of complex electrical circuits...

line 201: In the text, only Simulated Annealing was presented before. Derivative-free and Quasi-Newton were not explained.

line 231 & figure 4: Even though the resulting residuals of animal 2 are not significantly different, a clear difference is described, therefore it is difficult to refer it as a close performance. A figure showing the residuals of animal 2 is missing (analogous to figure 4b). Why are the residuals of animal 2 so much larger than for animal 1? Can the results of both animals (or more) be considered together? Then the groups would be larger (n=5 is very small).

line 202: How and according to which criteria is a TF classified as biologically inconsistent?

line 212: What is the difference between a run and an iteration?

line 246: The word ‘some’ overstates: Only the influence of the number of iterations and runs is investigated.

Missing: Demands on the test data: What is the minimum number of measuring points that input and output require for Iliski to run reliably?

Minor:

line 35: ‘addressed’ not ‘address’

figure 5: 'Yes' & 'no' is missing at the box 'are the signals noisy?'

Reviewer #2: This manuscript that describes use of a relatively simple platform to estimate transfer functions between inputs and outputs. The work is considered interesting in that it provides the public with a platform to estimate simple relationships in data. The platform is well presented and although I did not download and test, I am familiar with this sort of analysis. A few clarifications are requested prior to recommending publication.

Have the authors thought to include square (boxcar) functions as input? This is routinely done in fMRI data analysis. It would be useful to pair this with a different example of output data, especially data with different noise structure. For example, a prior publication from this group has ultrasound imaging data, that would be informative and expand the appeal for this platform.

Does the result of the optimization by correlation get a final amplitude adjustment by linear projection (least squares) or is it unit amplitude?

The authors should state whether this framework is able to handle complex-valued signals (sometimes used to maintain Gaussian noise). No need to modify the platform, just state for clarity.

Please clarify whether the TF dimension is the same as that of the input or it if is augmented by interpolation in some step.

Minor

Line45-46, this sentences are confusing since TFs simplify relationships into potentially single relationships. This needs to be clarified or modified.

Line195-96, Replace bonds with BOUNDS.

Line164, TF should not be italicized (that would mean variable T times F(t)), make it non-italicized to define it as one variable. It would be generally useful to also list TF(t) as TF(t; p1, p2, p3, p4) to make it clearer that the goal is to fit/manipulate those arguments.

**Have all data underlying the figures and results presented in the manuscript been provided?**

Reviewer #1: Yes

Reviewer #2: Yes

PLOS authors have the option to publish the peer review history of their article (what does this mean?). If published, this will include your full peer review and any attached files.

Reviewer #1: No

Reviewer #2: No
---

## [Editor Report · Decision Letter 1]

18 May 2021

Dear Dr charpak,

We are pleased to inform you that your manuscript 'Iliski, a software for robust calculation of transfer functions' has been provisionally accepted for publication in PLOS Computational Biology.

Best regards,

Dina Schneidman-Duhovny

Software Editor

PLOS Computational Biology

Dina Schneidman-Duhovny

Software Editor

PLOS Computational Biology

---

## [Editor Report · Acceptance letter]

8 Jun 2021

PCOMPBIOL-D-20-02258R1 

Iliski, a software for robust calculation of transfer functions

Dear Dr charpak,

I am pleased to inform you that your manuscript has been formally accepted for publication in PLOS Computational Biology. Your manuscript is now with our production department and you will be notified of the publication date in due course.

With kind regards,

Katalin Szabo
